# The Proteomic Landscape of Resting and Activated CD4+ T Cells Reveal Insights into Cell Differentiation and Function

**DOI:** 10.3390/ijms22010275

**Published:** 2020-12-29

**Authors:** Yashwanth Subbannayya, Markus Haug, Sneha M. Pinto, Varshasnata Mohanty, Hany Zakaria Meås, Trude Helen Flo, T.S. Keshava Prasad, Richard K. Kandasamy

**Affiliations:** 1Centre of Molecular Inflammation Research (CEMIR), Department of Clinical and Molecular Medicine (IKOM), Norwegian University of Science and Technology, 7491 Trondheim, Norway; yashwanth.subbannayya@ntnu.no (Y.S.); markus.haug@ntnu.no (M.H.); sneha.pinto@ntnu.no (S.M.P.); hany.z.meas@ntnu.no (H.Z.M.); trude.flo@ntnu.no (T.H.F.); 2Center for Systems Biology and Molecular Medicine, Yenepoya (Deemed to be University), Mangalore 575018, India; varsham@yenepoya.edu.in (V.M.); keshav@yenepoya.edu.in (T.S.K.P.)

**Keywords:** adaptive immunity, T-lymphocytes, CD4+ T helper cells, mass spectrometry, proteomics, label-free quantitation, systems biology

## Abstract

CD4+ T cells (T helper cells) are cytokine-producing adaptive immune cells that activate or regulate the responses of various immune cells. The activation and functional status of CD4+ T cells is important for adequate responses to pathogen infections but has also been associated with auto-immune disorders and survival in several cancers. In the current study, we carried out a label-free high-resolution FTMS-based proteomic profiling of resting and T cell receptor-activated (72 h) primary human CD4+ T cells from peripheral blood of healthy donors as well as SUP-T1 cells. We identified 5237 proteins, of which significant alterations in the levels of 1119 proteins were observed between resting and activated CD4+ T cells. In addition to identifying several known T-cell activation-related processes altered expression of several stimulatory/inhibitory immune checkpoint markers between resting and activated CD4+ T cells were observed. Network analysis further revealed several known and novel regulatory hubs of CD4+ T cell activation, including IFNG, IRF1, FOXP3, AURKA, and RIOK2. Comparison of primary CD4+ T cell proteomic profiles with human lymphoblastic cell lines revealed a substantial overlap, while comparison with mouse CD+ T cell data suggested interspecies proteomic differences. The current dataset will serve as a valuable resource to the scientific community to compare and analyze the CD4+ proteome.

## 1. Introduction

CD4+ T cells, also known as CD4+ T helper cells, are a subtype of T-lymphocytes that perform important immunoregulatory roles in adaptive immunity, including activation of B cells, cytotoxic T-cells, and nonimmune cells [1]. Bone marrow-derived hematopoietic progenitors committed of the T-cell lineage, unlike other cells of the hematopoietic cell lineage, enter circulation and migrate to the thymus where they undergo maturation and selection processes to produce a pool of mature resting CD4+ and CD8+ T cell lineages [2]. During maturation, each T cell re-arranges a unique T cell receptor (TCR) that recognizes specific antigenic peptides in complex with major histocompatibility complex (MHC) molecules on APCs. Mature naïve T cells exit the thymus and patrol the blood and lymph system where they screen MHC molecules on antigen-presenting cells (APCs) and are activated if their TCRs detect their cognate antigen on MHC molecules [3]. In addition, subsequent signals, including costimulatory receptor signaling (e.g., CD28) and environmental impacts such as cytokines, are essential for T cell activation [4]. The interactions between APCs and T cells are mediated by regulatory molecules known as immune checkpoints [5]. After activation, CD4+ T cells undergo clonal expansion and differentiation into CD4+ effector T cells. During this phase, cytokine signals from the environment impact the transcriptional programs in the activated CD4+ T cells and guide differentiation towards CD4+ T cell lineages which produce specific sets of effector cytokines. In 1986, Mosmann and colleagues identified two distinct types of CD4+ T helper cells: Type1 T helper cells (Th1) that produced IL-2, IL-3, IFNγ, and GM-CSF, and Type2 T helper cells (Th2) that produced IL-3, BSF1 (IL-4), a mast cell growth factor (IL-10) and a T cell growth factor [6]. However, over the years, multiple/a variety of subsets or lineages of CD4+ T helper cells have been identified depending upon their signature cytokine secretion, which includes—Th1, Th2, Th17, Th9, Th22, regulatory T cells (Tregs), and follicular helper T cells (TFH) [7,8]. Distinct transcriptional profiles and master transcriptional regulators have been identified for the different subsets/regulating lineage differentiation [9,10]. It is now understood that not all activated CD4+ T helper cells terminally differentiate, but that a substantial portion remains plastic and may be capable of acquiring other properties and functions as part of secondary immune responses [7].

CD4+ T cells play critical roles in the pathogenesis of several diseases, including infectious, auto-immune, inflammatory diseases, and malignancies. CD4+ T cells have a crucial role in the development of HIV infection, where virus entry into cells requires CD4 receptor involvement [11,12]. Progressive depletion of CD4+ T cell populations is one of the hallmarks of acquired immunodeficiency syndrome (AIDS) pathogenesis [13] resulting in increased susceptibility to opportunistic infections and virus-associated malignancies [14]. Interestingly, an increase in various CD4+ T cell subsets serves as a hallmark of inflammatory diseases such as multiple sclerosis, arthritis, allergies, and chronic airway inflammation in asthma [15]. Infiltration and accumulation of CD4+ T cells in the peripheral joints is an important feature of rheumatoid arthritis [16]. Additionally, CD4+ T cells play an important role in mediating crosstalk between immune cells and adipose tissues with an increase in adipose tissues known to be associated with obesity and obesity-associated diseases, including type 2 diabetes, insulin resistance, atherosclerosis, and stroke [17]. Increasing evidence now suggests a vital role of CD4+ T cells in tumor protection [18], driving several anti-tumor mechanisms [19,20,21,22]. Furthermore, CD4+ T cells have been shown to mediate direct cytotoxicity against tumor cells through increased production of interferon gamma (IFNγ) and tumor necrosis factor (TNFα in both preclinical models [23,24,25] and patient-derived CD4+ T cells [26]. CD4+ T cells can also induce humoral responses against tumor antigens primarily through increased expression of CD40 ligand that promotes differentiation and maturation of B-cells into affinity-matured, class-switched plasma cells [27,28].

T cell activation is accompanied with changes in transcriptional and proteomic machinery, including massive shifts in metabolism and biosynthesis which drives increase in size, rapid proliferation and differentiation of T cells [29]. To date, there have been several OMICs-based studies, specifically proteomics approaches to characterize changes in protein expression upon T cell activation [30,31,32,33,34,35]. Despite CD4+ T cells studied extensively, several aspects pertaining to T cell biology, such as comprehensive knowledge of the proteomic repertoire, signaling mechanisms, patterns of heterogeneity in the population, interspecies differences, and fundamental differences between primary cells and cell line models, are not well characterized. The aim of this study was to add knowledge to the proteomic differences between resting and in vitro activated primary human CD4+ T cells as well as common laboratory-used model T cell lines. We carried out label-free comparative proteomic analysis of resting (unactivated) and TCR-activated (72 h) primary human CD4+ T cells purified from two healthy donors. Furthermore, we also profiled the proteome expression repertoire of the human T cell lymphoblastic cell line SUP-T1 and accessed a previously published proteome profile of Jurkat T lymphoblast cells [36]. Using an integrative bioinformatics approach, we compared the proteomic profiles of resting and activated primary CD4+ T cells with those of human SUP-T1 and Jurkat T lymphoblast cell lines. We also provide a comprehensive overview of signaling pathways and networks affected by the activation of CD4+ T cells. The data generated from the current study will enable us to gain a better understanding of the molecular machinery operating within primary CD4+ T cells during T cell activation, the proteomic differences between primary CD4+ T cells and model T cell lines as well as interspecies differences between human and mice.

## 2. Results

### 2.1. Comparative Proteomic Analysis of Resting and Activated Primary CD4+ T Cells and SUP-T1 T Lymphoblastic Cell Line

We performed an unbiased global proteomic profiling to elucidate the protein expression profiles of resting (un-activated) and TCR- activated primary human CD4+ T cells (Figure 1A) as well as the T cell lymphoblastic cell line SUP-T1. Peripheral Blood Mononuclear Cells (PBMCs) were isolated from two healthy donors, and CD4+ T cells were further purified using magnetic bead-based isolation (negative-isolation of “untouched” cells). Flow-cytometric analysis indicated a purity of >94% for both donors and in total less than 0.5% contaminating of myeloid origin (CD14+ or CD11c+ cells) or CD8+ T cells (Figure 1B,C). Resting CD4+ T cells were frozen directly after isolation. For activated CD4+ T cell samples, TCR (plate-bound anti-CD3 antibody) and co-stimulatory (anti-CD28 antibody) signaling was induced for 72 h before the samples were frozen.

LC-MS/MS analysis of resting and activated CD4+ T cell proteomes resulted in the identification of 5237 proteins (Appendix A). The proteomic profiles of resting and activated CD4+ T cells from two donors were compared to see expression patterns using Principal Component Analysis (PCA). While resting cells from the donors clustered together in the PCA plot, activated cells revealed heterogeneous expression profiles between the two donors (Figure 1D), indicating that there were significant proteomic differences between them. A total of 1119 were significantly altered between resting and activated states in both donors (log2 fold change ±2, *p*-value < 0.05) (Appendix A; Appendix A). The top 10 differentially regulated proteins in activated CD4+ T cells include enzymes such as thymidylate synthetase (TYMS) and methylenetetrahydrofolate dehydrogenase (NADP+ dependent) 2 (MTHFD2) that were found to upregulated while ATP synthase membrane subunit 6.8PL (ATP5MPL), azurocidin 1 (AZU1), granzyme K (GZMK) were found to be downregulated in activated CD4+ T cells from both the donors (Figure 1E,F).

### 2.2. Known and Novel Molecular Markers of Resting and Activated CD4+ T Cells

We analyzed proteins characteristic of resting and activated CD4+ T cells. As expected, CD4 was uniformly expressed in both resting and activated CD4+ T cells (Figure 2A). CD8 protein was not identified in the current dataset, confirming the purity of the CD4+ T cell preparations. The expression profiles of the hallmark markers of T cell activation including transcriptional regulator FOXP3, interleukin (IL)-2 receptor α-chain IL2RA (CD25), major histocompatibility complex class II, DR alpha (HLA-DRA), CD40 ligand (CD40LG) and CD69 were assessed. As expected, a marked increase in the protein expression was observed after 72 h of activation in comparison to resting CD4+ T cells. HLA-DRA however, showed a minor increase in protein expression (Figure 2B–F).

Next, we carried out gene ontology-based enrichment analysis of differentially expressed proteins in activated CD4+ T cells with respect to their resting/untreated counterparts. The two donors showed significant differences in the type of biological processes for which proteins were upregulated in CD4+ T cells after activation (Figure 2G,H). However, “housekeeping” processes pertaining to cell cycle, mitosis and DNA replication were commonly enhanced in both donors. The biological processes enriched for downregulated proteins were mostly similar for both donors and consisted mainly of proteins involved in neutrophil activation (Appendix A).

Furthermore, a detailed comparison of proteins differentially expressed in activated CD4+ T cells compared to resting CD4+ T cells was carried out. Gene sets derived from the Molecular Signatures Database (MSigDB) were used to obtain insights into signaling in the course of T cell activation and modulating effector lineage differentiation. Proteins involved in T cell activation, mainly including transcription factors (basic leucine zipper ATF-like transcription factor (BATF), forkhead box P3 protein (FOXP3), T-box transcription factor 21 (TBX21/T-Bet), interferon regulatory factor 4 (IRF4), among others were found to be overexpressed in activated cells of both donors. On the contrary, proteins, including BCL3 transcription coactivator (BCL3), zinc finger, and BTB domain containing 7B (ZBTB7B) revealed mixed patterns of expression between the two donors. Notably, the expression of SLAMF6 was decreased after activation (Figure 2I). Among the regulators of T-cell activation, we observed proteins belonging to the CD family (CD47, CD74, CD81, CD70, and CD40LG), members of TNF receptor superfamily (TNFRSF1B, TNFRSF18), interferon regulatory factor 4 (IRF4) and inducible T cell costimulator (ICOS) to be overexpressed in activated CD4+ T cells from both donors (Figure 2J).On the contrary, interleukin 7 receptor (IL7R), CD300A, spleen associated tyrosine kinase (SYK), and NFKB activating protein (NKAP) were decreased in activated CD4+ T cells compared to their resting counterparts. Furthermore, we assessed the expression of Th cell lineage markers which suggested similar expression patterns across the donors with a mixed lineage phenotype observed in both donors (Figure 2K). Notably, three proteins constituting Th1 cell hallmark cytokine Interferon-γ (IFNG), signal transducer, and activator of transcription 4 (STAT4) and (T-bet/TBX21) and Treg markers -transcription factors STAT5A, STAT5B, and FOXP3 and TGFB1 effector cytokine were increased after activation were increased consistently in both donors after activation.

### 2.3. Activation of CD4+ T Cells Influences Processes and Signaling Pathways

Kinases and phosphatases constitute important classes of proteins mediating cell signaling and could provide mechanistic insights into T cell activation. Towards this end, we explore the expression profile of protein kinases and phosphatases in resting and activated CD4+ T cells (Figure 3A,B). Protein kinases including Aurora kinase B (AURKB), cyclin dependent kinases (CDK1 and CDK2), RIO kinases (RIOK1 and RIOK2), checkpoint kinase 1 (CHEK1), Janus kinase 3 (JAK3), serine/threonine kinase 17b (STK17B) and SRSF protein kinase 1 (SRPK1) were significantly upregulated (*p*-value < 0.05) 72 h post-activation in CD4+ T cells from both donors. On the contrary, cyclin dependent kinase 13 (CDK13), TBC1 domain containing kinase (TBCK), members of the MAP kinase family (MAPKAPK3, MAP3K2, and MAPK9), microtubule affinity regulating kinase 3 (MARK3), SNF related kinase (SNRK), ribonuclease L (RNASEL) and spleen associated tyrosine kinase (SYK) were significantly downregulated in activated CD4+ T cells from both donors. In general, more protein kinases were upregulated in activated CD4+ T cells from donor 1 compared to donor 2. Protein kinases such as mitogen-activated protein kinase 8 (MAPK8), calcium/calmodulin-dependent protein kinase ID (CAMK1D), mitogen-activated protein kinase 6 (MAPK6), PEAK1 related, kinase-activating pseudokinase 1 (PRAG1), TRAF2 and NCK interacting kinase (TNIK), misshapen like kinase 1 (MINK1) and ribosomal protein S6 kinase A4 (RPS6KA4) were only found to be upregulated in activated CD4+ T cells from donor 1 suggesting better activation of these cells. Among the phosphatases, the non-protein phosphatase- myotubularin related protein 9 (MTMR9) was found to be upregulated in activated CD4+ T cells, whereas the protein tyrosine phosphatase receptor type A (PTPRA) was found to be downregulated in activated CD4+ T cells from both donors. A significant decrease in the expression of protein tyrosine phosphatase receptor type J (PTPRJ), myotubularin related protein 12 (MTMR12), and synaptojanin 2 (SYNJ2) was observed in activated CD4+ T cells from donor 2. These suggest that phosphatases were less impacted after activation in both donors.

Pathway enrichment analysis of proteins differentially expressed in activated CD4+ T cells from both the donors revealed several pathways, such as Interferon signaling, cell cycle pathways, and nucleotide metabolism, to be enriched in activated CD4+ T cells (Figure 3C). This is in line with reports suggesting increased cytokine-mediated differentiation and increased proliferation of activated CD4+ T cells. We further assessed the expression of proteins involved in adaptive immune response, including cytokines, cytokine receptors, hypoxia and ROS markers (Appendix A). We observed an overall increase in the extent of expression of cytokines and receptors in activated CD4+ T cells with the exception of interleukin 7 receptor (IL7R) which was downregulated. Protein markers of hypoxia, including perilipin 2 (PLIN2), high density lipoprotein binding protein (HDLBP), jumonji domain containing 6, arginine demethylase and lysine hydroxylase (JMJD6), hexokinase 2 (HK2), ilvB acetolactate synthase like (ILVBL), solute carrier family 2 member 1 (SLC2A1), and prolyl 4-hydroxylase subunit alpha 1 (P4HA1), were found to be upregulated in activated CD4+ T cells of both donors. Further, ROS markers apolipoprotein E (APOE) was found to be increased in expression in activated T cells while neutrophil cytosolic factor 2 (NCF2), ring finger protein 7 (RNF7), myeloperoxidase (MPO), and thioredoxin reductase 2 (TXNRD2) were found to be decreased. In summary, activated CD4+ T cells showed classical markers of adaptive immune responses, while the increased hypoxia markers in response to activation may be a result of increased nutritional needs of activated CD4+ T cells. In terms of cellular metabolism, proteins/enzymes involved in amino acid and lipid metabolism were in general upregulated after activation, whereas a mixed expression pattern between resting and activated CD4+ T cells were observed for oxidative phosphorylation and glycolysis/gluconeogenesis (Appendix A). Notably, acyl-CoA oxidase 1 (ACOX1), holocytochrome c synthase (HCCS), 3-hydroxy-3-methylglutaryl-CoA synthase 1 (HMGCS1), ELOVL fatty acid elongase 5 (ELOVL5), retinol saturase (RETSAT), isopentenyl-diphosphate delta isomerase (IDI1), fatty acid synthase (FASN) corresponding to lipid metabolism and folylpolyglutamate synthase (FPGS), methionyl-tRNA synthetase 2, mitochondrial (MARS2), fumarylacetoacetate hydrolase (FAH), and tryptophanyl-tRNA synthetase 1 (WARS), corresponding to amino acid metabolism were found to be upregulated in activated CD4+ T cells. Several proteins belonging to cellular processes such as cell cycle, apoptosis, autophagy, and phagocytosis were upregulated in activated CD4+ T cells indicating changes in cellular activity and cell proliferation in response to activation (Appendix A).

### 2.4. Activation of CD4+ T Cells Influences Protein Signaling Networks

Interactome analysis of proteins upregulated during activation of primary human CD4+ T cells was carried out on our datasets to identify critical regulatory hubs. The protein-protein interaction network was generated from the set of proteins upregulated in activated CD4+ T cells from both donors compared to non-activated CD4+ T cells using Cytoscape, and network topology parameters were calculated (Figure 4, Appendix A). The betweenness centrality and degree measures were used to visualize the network and identify the highly connected nodes. Proteins with high betweenness centrality measures included proteins with known roles in T cell activation such as transcription factors- interferon regulatory factor 1 (IRF1), forkhead box P3 (FOXP3), Interferon-γ (IFNG), as well as marker of proliferation Ki-67 (MKI67), cell cycle proteins including cyclin dependent kinase 1 (CDK1), cyclin dependent kinase 2 (CDK2), cell cycle associated protein 1 (CAPRIN1) and aurora kinase B (AURKB). Interestingly several proteins not previously known in the context of T cell activation including RIO kinase 2 (RIOK2), cell division cycle 20 (CDC20), tRNA methyltransferase 1 (TRMT1), ubiquitin conjugating enzyme E2 L6 (UBE2L6), clusterin (CLU) and mini-chromosome maintenance proteins (MCM) including MCM2, MCM3, MCM4, MCM5, MCM5, and MCM7 also showed a significant and high betweenness centrality measures probably representing novel regulatory hubs of CD4+ T cell activation and need to be investigated further.

### 2.5. Changes in Expression Levels of Immune Checkpoint Proteins in Activated/Resting CD4+ T Cells

The complete activation of CD4+ T cells depends on signals from the TCR (signal 1) and antigen-independent co-stimulatory signals (signal 2). In our set-up, we provided both signal 1 (anti-CD3 stimulation) and signal 2 (anti-CD28 stimulation) to primary CD4+ T cells for 72 h. However, CD4+ T cell activation is tightly regulated by a range of co-stimulatory and co-inhibitory signaling receptors known as immune checkpoints. Immune checkpoints act as regulators of the immune system mediating interactions between T cells and other cells, such as APCs or tumor cells [5]. These inhibitory or stimulatory checkpoint pathways attenuate T cell activation and are essential for self-tolerance mechanisms and regulate the adaptive immune response [37,38]. Immune checkpoints are considered as important immunotherapy targets, and checkpoint inhibitors against CTLA4, PD-1/PD-L1 have been approved for clinical use [39]. Of the 16 known immune checkpoints [38,40], ten were identified in the current dataset (Figure 5). These included stimulatory checkpoint molecules such as CD27 (TNFRSF7), CD28, CD40LG (CD40L), CD96 (Part of TIGIT/CD96), inducible T cell costimulator (ICOS), TNF receptor superfamily members-TNFRSF4/OX40 and TNFRSF18/GITR. Among the inhibitory checkpoint molecules, cytotoxic T-lymphocyte associated protein 4 (CTLA4), lymphocyte activating 3 (LAG3), and V-set immunoregulatory receptor (VSIR/VISTA/PD-1H) were identified.

Amongst these, CD27 (TNFRSF7), CD28, and VSIR (VISTA/PD-1H) proteins were found to be expressed at similar levels in both resting and activated CD4+ T cells, CD96 was expressed at higher levels in resting compared to activated CD4+ T cells. Interestingly, PD-1 (PDCD1) was neither detected in resting nor activated CD4+ T cells in the current study. This may be because PD-1 is generally strongly expressed in late-stage exhausted T cells, such as those in the tumor environment and was not detected in our setup with strongly activated T cells at an early stage (72 h). However, expression of the stimulatory checkpoint molecules CD40LG, ICOS, TNFRSF18 and TNFRSF4 (OX40) as well as the inhibitory immune checkpoint regulators CTLA4 and LAG3 was significantly upregulated 72 h post-activation in CD4+ T cells. These results indicate that CD4+ T cells are highly responsive for distinct positive co-stimulatory signals 72 h post-activation, but that at the same time, expression of certain inhibitory checkpoint molecules is induced and counter-acts the activation process.

### 2.6. Comparison of Primary T Cell Data with Proteomes of Human Lymphoblastic T Cell Lines and Published Primary Human and Mouse CD4+ T Cell Datasets

In vitro cell models such as SUP-T1 and Jurkat cells are widely employed to study T-cell mediated signaling mechanisms and are in general amenable to gene manipulation techniques. However, there is a paucity of information on the extent of correlation of protein expression in SUP-T1 cells with protein expression in primary CD4+ T cells. To achieve this, we generated the proteome profile of SUP-T1 cells. As these cells do not express a functional TCR on their surface, they were analyzed in a resting state (no TCR activation prior to analysis). LC-MS/MS analysis resulted in the generation of 167,863 peptide spectrum matches (PSMs), which mapped 26,111 peptides corresponding to 4815 proteins (Appendix A). Comparison of the proteomic profiles of SUP-T1 cells and Jurkat A 3 cells obtained from a published dataset [36] with the data from resting primary CD4+ T cells revealed 2925 common proteins (~59%) (Figure 6A, Appendix A). Gene ontology-based enrichment of this subset revealed housekeeping processes such as RNA processing, translation initiation, and nuclear transport. We further assessed gene ontology-based enrichment for the proteins that were exclusive to each of the cell types. The top enriched processes unique to resting primary CD4+ T cells included proteins involved in interferon gamma-mediated signaling pathway and T cell activation processes, suggesting that these processes were less represented in SUP-T1 and Jurkat cells. CD4 protein was found to be expressed in all cell types. Further examination into CD3 (a T cell co-receptor) expression revealed that CD3D and CD3E subunits were found to be expressed in all three cell types, while CD3G was found to be expressed in primary resting T cells and Jurkat but not SUP-T1 cells. However, CD28 expression was identified in resting and SUP-T1 cells, but not Jurkat cells (Appendix A). In addition to CD4, SUP-T1 cells also expressed CD8 (CD8B) albeit at lower levels. These findings suggest that there is a difference between T cell lines which may affect the phenotype under study.

To assess the extent of similarities/dissimilarities across published studies on human and mouse CD4 T cell datases, we performed a series of comparisons across datasets. Although a substantial overlap was observed between proteomic profiles of human resting and activated CD4+ T cells with a previous study on proteomic analysis of mouse resting CD4+ T cells and Th1 cells [41] (Figure 6B, Appendix A), a closer inspection of the pattern of expression demonstrated a poor correlation. This strongly indicates considerable species-level differences in the protein expression profiles of both resting and activated CD4+ T cells (Figure 6C).

Similar findings were also observed upon comparison of our dataset with previously published proteomic datasets on human CD4+ T cells [30,42,43] (Appendix A). A large overlap of identified proteins was observed across both resting and activated CD4+ T cells irrespective of the generation of mass spectrometers used (Figure 6D,E). However, correlation matrices indicated a considerable heterogeneity in protein expression profiles between the different datasets. Overall, the proteomic profile provided by Rieckmann et al. [43] showed more similarity to the current study, while the proteome profiles from Gerner et al. [30] and Mitchell et al. [42] showed lower levels of correlation (Figure 6F). Taken together, our analysis demonstrates heterogeneity across individuals and the technology employed strongly affect the proteome dynamics of T-cells.

## 3. Discussion

Primary human CD4+ T cells have been studied extensively pertaining to their role in adaptive immunity. Several omics-based studies have characterized the proteome of CD4+ T cells. Despite these cells being studied in detail, the protein expression dynamics as well as signaling mechanisms operating in these cells during steady-state and upon induction of activation is not entirely understood. To identify changes in the protein expression profiles of resting and TCR-activated CD4+ T cells, we performed a label-free quantitative proteomic analysis on primary human CD4+ T cells derived from two donors. We also carried out the proteomic analysis of SUP-T1 cells, an in vitro T cell model cell line widely used in the field to compare and contrast proteome profiles of primary CD4+ T cells. Our study indicated that during the course of CD4+ T cell activation, significant changes in the protein expression profile occur. We identified several proteins that were previously found to be differentially expressed in response to stimulation. Importantly, markers of CD4+ T cell activation along with their regulators and Th1-, Th2-, Th17-, and Treg-specific markers were found to be upregulated, suggesting the presence of potential transient hybrid cell types. This has been previously suggested by logical modeling-based simulations on T cell differentiation [44]. These findings suggest heterogeneity in CD4+ T helper cell types during activation and differentiation into terminally differentiated CD4+ T effector cells. This heterogeneity may arise due to varying stimuli that the donors were exposed to throughout life, the in vitro condition of cells, the single timepoint used in the study-72 h, which probably represents an intermediate phase where the T cells may not be terminally differentiated into effector lineages. Further, the in vitro conditions used may not mimic the in vivo state where polarizing cytokine signals would usually arise from APCs.

Further, several protein kinases, cytokines, MAP kinases, and markers of adaptive immune response, ROS, and hypoxia were found to be upregulated. Interestingly, proteins belonging to lipid metabolism and amino acid metabolism were found to be upregulated in activated CD4+ T cells. This can be possibly be explained by the increasing cellular metabolic needs due to CD4+ T cell activation and proliferation. In addition, several proteins involved in amino acid synthesis and transport were upregulated in activated CD4+ T cells, indicating that uptake and utilization of nutrients are associated with T cell differentiation and function. Our findings are in concordance with previous studies suggesting amino acid transporters are essential for the normal differentiation and functioning of T cells [45,46,47,48,49]. Previous reports have shown that CD4+ T cell activation results in substantial remodeling of the mitochondrial proteome that, in turn, generates specialized mitochondria with significant induction of the one-carbon metabolism pathway [50]. Defective one-carbon metabolism has been shown to result in defective resting T cell activation in aged mice [51]. In the current study, we identified several proteins belonging to the One carbon pool by folate pathway induced in CD4+ T cells after activation, including dihydrofolate reductase (DHFR), methylenetetrahydrofolate dehydrogenase (NADP+ dependent) 1 like (MTHFD1L), methylenetetrahydrofolate dehydrogenase (NADP+ dependent) 2, methenyltetrahydrofolate cyclohydrolase (MTHFD2), and thymidylate synthetase (TYMS). This confirms the previous findings of mitochondrial remodeling after CD4+ T cell activation.

An additional objective of this study was to compare proteomes between primary CD4+ T cells and commonly used T cell lines including Jurkat and SUP-T1 to see if the biological mechanisms in these cell models mimic the primary CD4+ T cells. Comparison of the lists of identified proteins across the three cell types showed the majority of the proteome expressed in these cells was common, driving essential processes of transcription, translation, and transport. However, each of the cells also expressed proteomes exclusive to each cell type, which might be caused due to varying proteomic coverage and depth in the different experiments. Differences were observed in terms of expression of T cell activators -CD3 and CD28 across these cell types, with CD3G not being identified in SUP-T1. This is in accordance with previous reports showing SUP-T1 cells to be CD3-negative [52,53]. It is widely known that Jurkat cell lines are CD28+, and it not being detected in the previous proteomics data suggests low proteome coverage as a possible cause. Both SUP-T1 and Jurkat cell lines exhibited enrichment of processes such as double-strand break repair, histone modification, cell junction organizations, which could be due to the malignant nature of these lymphoblastic cell lines. In comparison, primary CD4+ T cells showed lower enrichment of repair processes. It can be concluded that these cell lines are essentially similar in terms of shared proteomes and can serve as useful models of resting primary CD4+ T cells. However, prior knowledge of the proteomes of these cell lines is desirable to study specific biological processes. Comparison of our data on CD4+ T cells with proteomic data from mouse CD4+ T cells showed lower levels of correlation, suggesting interspecies differences in CD4+ T cell activation, which may lead to poor translatability of findings of mouse-based experiments to the physiological state of T cell activation in humans. However, this needs to be validated and explored further.

We also identified heterogeneity in the levels of CD4+ T cell activation markers present in the cells from two donors. In our study, cells from donor 1 were observed to show more markers of activation, suggesting better activation. Further, resting cells from Donor 1 showed higher levels of protein kinases and phosphatases. This could be attributed to a higher degree of variability among primary donors with respect to their age, immune status, and recent exposure to infection. Comparison of CD4+ T cell proteomic profiles with other previously published studies also indicated considerable heterogeneity in proteomic expression profiles of both resting and activated T cells. The heterogeneity also extended to important regulatory molecules of the adaptive immune response, such as immune checkpoints.

We identified several potential regulatory hubs of CD4+ T cell activation using network analysis. These included proteins involved in inflammatory response, including VTN; CD40LG; IFNG; IL2RA; FN1; IL2RG; TNFRSF1B. Interestingly, several proteins associated with the cell cycle, including CDK1, CDK2, CAPRIN1, CCNB1, and CDC20, are likely to be potentially regulatory hubs of CD4+ T cell activation. Previous studies have suggested that cell cycle progression and cytokine signaling are closely linked during CD4+ effector T cell differentiation [54,55,56,57,58,59]. The upregulation of proteins involved in DNA replication such as MCM2-7 POLD3, POLA1, PCNA, and PRIM1; pyrimidine metabolism proteins such as RRM1; UCK2; RRM2; TK1; TYMS; TYMP; and the marker KI67 (MKI67) during activation is explained the increased rate of cell proliferation. Protein kinases such as RIOK2, AURKB, and PRKAB1 were found to be significantly upregulated in our datasets after activation of CD4+ T cells. Among these, RIOK2 and AURKB were found to have potential roles as regulatory hubs from the network analysis. Both RIOK2 and AURKB are associated with cell cycle activities. Aurora kinase B (AURKB) has been shown to regulate CD28-dependent T cell activation and proliferation [60]. RIO kinase 2 (RIOK2) is one of the members of the atypical protein kinase families [61]. RIOK2 is poorly studied compared to the other kinases; therefore, the biological mechanisms mediated by it are not well known. Over the years, a few papers have explored the role of RIOK2 in the context of the cell cycle. A study by Read and colleagues investigated a Drosophila glioblastoma model and discovered the Akt-dependent overexpression of RIOK1 and RIOK2 in glioblastoma cells [62]. Further, the study also found that the decreased expression of these kinases caused aberrant Akt signaling and resulted in cell cycle and apoptosis [62]. A recent study identified RIOK2 silencing in glioma cells inhibited cell migration and invasion [63]. RIOK2 has also been found to be essential for ribosome biogenesis [64,65], which in turn is regulated in a cell cycle-dependent fashion [66]. However, there are no previous reports of RIOK2 being associated with T cell activation. In a previous paper, it was shown that PLK1 phosphorylates and activates RIOK2 and this in turn leads to mitotic progression [67]. Polo-like kinase 1 (PLK1) has well-known roles in T cell function [68,69]. We thus hypothesize that RIOK2 might be involved in the division and function of T cells function. While the modulatory role of several of these potential regulatory hubs in T cell activation is well known, others such as RIOK2 need to be studied further.

## 4. Materials and Methods

### 4.1. Cells

Buffy coats from healthy blood donors were received from the Blood Bank (St Olav’s Hospital, Trondheim) with approval by the Regional Committee for Medical and Health Research Ethics (REC Central, Norway, NO. 2009/2245). Peripheral Blood Mononuclear Cells (PBMCs) were isolated from Buffy coats by density gradient centrifugation (Lymphoprep, Axis-shield PoC AS, Oslo, Norway). CD4+ T cells were isolated from PBMCs by a magnetic bead “negative” isolation procedure using the CD4+ T Cell Isolation Kit (Miltenyi Biotec, Bergisch Gladbach, Germany) and LS columns (Miltenyi Biotec, Bergisch Gladbach, Germany). CD4+ T cell purity was assessed by flow cytometry using anti-CD4 Alexa 700 (eBioscience, San Diego, CA, USA) and anti-CD3 Brilliant Violet (BV) 785 (BioLegend, San Diego, CA, USA) antibody staining. Data were acquired on a BD LSRII flow cytometer and analyzed using FlowJo software (FlowJo, LLC, Ashland, OR, USA). For both donors, CD4+ T cell purity was >94% and cell preparations contained less than 0.1% CD8+ cells and less than 0.5% cells CD11c+ or CD14+ cells of myeloid origin (Figure 1B,C). SUP-T1 human T lymphoblast cells (ATCC) were cultured in RPMI 1640 (Gibco, Dublin, Ireland) supplemented with 10% FBS and penicillin/streptomycin (Thermo Fisher Scientific, Rockford, IL, USA).

### 4.2. CD4+ T Cell Activation

For the unactivated (resting) CD4+ T cell samples, 1 × 10^7^ CD4+ T cells from both donors were washed three times with PBS before the pellet was shock-frozen in liquid nitrogen and stored at −80 °C. For the activated CD4+ T cell samples, 1 × 10^7^ CD4+ T cells from both donors were activated in anti-CD3 coated plates (clone OKT3, eBioscience, 5 µg/mL, 1 h) in the presence of 1 μg/mL anti-CD28 (clone CD28.2, eBioscience). CD4+ T cells were cultured for 72 h in RPMI 1640 (Sigma-Aldrich, Darmstadt, Germany), supplemented with 10% pooled human serum (The Blood Bank, St Olav’s Hospital, Trondheim, Norway) at 37 °C and 5% CO_2_. CD4+ T cells were washed three times with PBS before the pellet was shock-frozen in liquid nitrogen and stored at −80 °C.

### 4.3. Sample Preparation of CD4+ T Cell for Proteomics

The cell lysates were reconstituted in 300 µL lysis buffer containing 4% sodium dodecyl sulfate (SDS) and 50 mM triethyl ammonium bicarbonate (TEABC) (Sigma-Aldrich). They were sonicated three times for 10 s on ice, followed by heating at 90 °C for 5 min. The lysate was further centrifuged at 12,000 rpm for 10 min. The concentration of protein was determined using bicinchoninic acid assay (BCA) (Thermo Fisher Scientific). The samples were subjected to in-solution trypsin digestion and subjected to strong cation exchange-based fractionation

Briefly, 200 µg of protein lysate of resting and activated CD4 were considered for trypsin digestion, where it was reduced by incubating in 10 mM dithiothreitol (DTT) (Sigma-Aldrich) at 60 °C for 20 min and alkylated using 20 mM iodoacetamide (IAA) Edited at room temperature for 10 min. This was followed by acetone precipitation for 6 h, where the pellet was dissolved in 50 mM TEABC. The lysate was then subjected for digestion using l-(tosylamido-2-phenyl) ethyl chloromethyl ketone (TPCK) treated trypsin (Worthington Biochemical Corporation, Lakewood, NJ, USA) at a final concentration of 1:20 (*w*/*w*) at 37 °C overnight (~16 h). SCX fractionation was carried out as described previously [70].

### 4.4. Tandem Mass Spectrometry (MS/MS) Analysis

The digested peptides were analyzed on Orbitrap Fusion Tribrid mass spectrometer (Thermo Scientific, Bremen, Germany) interfaced with Easy-nLC-1200 (Thermo Scientific, Bremen, Germany). Each fraction was reconstituted in 0.1% formic acid and loaded onto the trap column (75 µm × 2 cm, nanoViper, 3 µm, 100 A°) filled with C18 at a flow rate of 4 µL/min with Solvent A. The peptides were then resolved onto the analytical column (15 cm × 50 µm, nanoViper, 2 µm) for 120 min. Data were acquired by using data-dependent acquisition mode at a scan range of 400–1600, in positive mode with a maximum injection time of 55 ms using an Orbitrap mass analyzer at a mass resolution of 120,000. MS/MS analysis was carried out at a scan range of 400–1600. Top ten intense precursor ions were selected for each duty cycle and subjected to higher collision energy dissociation (HCD) with 35% normalized collision energy. The fragmented ions were detected using Orbitrap mass analyzer at a resolution of 120,000 with maximum injection time of 200 ms. Internal calibration was carried out using a lock mass option (*m*/*z* 445.1200025) from ambient air.

### 4.5. Bioinformatics Analysis of Mass Spectrometry Data

The raw data obtained from mass spectrometry analysis were searched against the human UniProt protein database (20,972 sequences, downloaded from ftp://ftp.uniprot.org/ on 3 July 2019) using MaxQuant (v1.6.10.43,) search algorithm. Trypsin was specified as the protease, and a maximum of two missed cleavages was specified. N-terminal protein acetylation and oxidation of methionine were set as variable modifications, while carbamidomethylation of cysteine was set as a fixed modification. The peptide length was set between 8–25 and precursor, and fragment mass tolerances were specified as 20 ppm each. Decoy database search was used to calculate False Discovery Rate (FDR), which was set to 1% at PSM, protein, and peptide levels. The search results from MaxQuant were processed and label-free protein quantitation using Perseus (v. 1.6.2.2, https://maxquant.net/perseus/) [71]. Briefly, intensity values were filtered, log-transformed, and fold-change calculations were performed. Perseus was also used to generate volcano and PCA plots.

Hypergeometric enrichment-based gene ontology and pathway analysis were carried out with R (R studio v. 1.2.1335, Bioconductor v 3.9.0) scripts using clusterProfiler (v. 3.12.0) [72] and Reactome pathways [73] with ReactomePA package (v. 1.28.0) [74]. The pathway enrichment parameters included 0.05 as *p*-value cut-off, Benjamini-Hochberg correction based *p*-value adjustment, minimum gene set size of 10, and *q*-value cut-off of 0.2. Pathways were plotted in R using ggplot2 package (v. 3.3.0, https://cran.r-project.org/web/packages/ggplot2/).

The Gene Ontology (GO) enrichment for Biological processes was carried out using R with clusterProfiler. The GO enrichment parameters included 0.05 as *p*-value cut-off, Benjamini-Hochberg correction based *p*-value adjustment, minimum gene set size of 10. Gene lists for functions such as Cell cycle, phagocytosis, autophagy, apoptosis, hypoxia, adaptive immune response, and T cell activation were obtained from the Molecular Signatures Database (MSigDB, v. 7.0, https://www.gsea-msigdb.org/gsea/msigdb) [75]. Gene lists for metabolism was obtained from KEGG (https://www.genome.jp/kegg/), while genes list for reactive oxygen species was compiled from the literature [76]. Protein kinase and phosphatase lists were obtained, as described previously [77]. Immune checkpoints receptors and their ligands were compiled from the literature [38,40] and compared with the data from this study. Heatmaps were drawn using Morpheus (https://software.broadinstitute.org/morpheus/) with Euclidean complete linkage-based hierarchical clustering. Networks were generated using StringApp [78] in Cytoscape (version 3.7.1) [79] as previously described [80]. Briefly, proteins that were upregulated in both donors and significant were filtered and used to generate networks. The network properties were calculated using NetworkAnalyzer in Cytoscape [81], and the network was visualized based on betweenness centrality and degree values.

### 4.6. Isolation of SUP-T1 Cell Proteome and MS/MS Analysis

The cell lysates of SUP-T1 was reconstituted in 300 µL lysis buffer containing 4% sodium dodecyl sulfate (SDS) and 50 mM triethyl ammonium bicarbonate (TEABC). It was sonicated three times for 10 s on ice, followed by heating at 90 °C for 5 min. The lysate was further centrifuged at 12,000 rpm for 10 min. The concentration of protein was determined using bicinchoninic acid assay (BCA), giving a yield of 11.3 µg/µL. Protein lysate of 200 µg was considered for trypsin digestion where it was reduced by incubating in 10 mM dithiothreitol (DTT) at 60 °C for 20 min and alkylated using 20 mM iodoacetamide (IAA) at room temperature for 10 min. This was followed by acetone precipitation for 6 h where the pellet was dissolved in 50 mM TEABC. The lysate was then subjected for digestion using L-(tosylamido-2-phenyl) ethyl chloromethyl ketone (TPCK) treated trypsin (Worthington Biochemical Corporation, Lakewood, NJ, USA) at a final concentration of 1:20 (*w*/*w*) at 37 °C overnight (~16 h). SCX fractionation was carried out as described previously [70]. The digested peptides were analyzed on Orbitrap Fusion Tribrid mass spectrometer (Thermo Scientific, Bremen, Germany) interfaced with Easy-nLC-1200 (Thermo Scientific, Bremen, Germany). Each fraction was reconstituted in 0.1% formic acid and loaded onto the trap column (75 µm × 2 cm, nanoViper, 3 um, 100A°) filled with C18. The peptides were then resolved onto the analytical column (15 cm × 50 µm, nanoViper, 2 µm) for 120 min at a flow rate of 250 nL/min. Data were acquired by using data-dependent acquisition mode at a scan range of 400–1600, in positive ion mode with a maximum injection time of 10 ms using an Orbitrap mass analyzer at a mass resolution of 120,000. MS/MS analysis was carried out at a scan range of 110–1800. MS/MS analysis was carried out in Top Speed mode, and the precursor ions were subjected to higher collision energy dissociation (HCD) with 33% normalized collision energy. The fragmented ions were detected using Orbitrap mass analyzer at a resolution of 30,000 with maximum injection time of 200 ms. Internal calibration was carried out using a lock mass option (*m*/*z* 445.1200025) from ambient air. Mass spectrometry derived data was searched against Human RefSeq 81 protein database in Proteome Discoverer 2.1 (Thermo Scientific, Bremen, Germany) using SEQUEST and Mascot (version 2.5.1, Matrix Science, London, UK) search algorithms. The parameters included trypsin as a proteolytic enzyme with maximum two missed cleavage where cysteine carbamidomethylation was specified as static modification and acetylation of protein N-terminus and oxidation of methionine was set as dynamic modifications. The length of 7 amino acids was set as the minimum peptide length. The search was carried out with a precursor mass tolerance of 10 ppm and fragment mass tolerance of 0.05 Da. The data were searched against the decoy database with a 1% FDR cut-off at the peptide level.

### 4.7. Comparison with Published Datasets

We carried out comparisons of the data from this study with previously published datasets to gain a better understanding of the proteomic landscapes of T cells. We downloaded protein expression datasets of published studies and mapped them to gene symbols using a combination of g:Profiler (https://biit.cs.ut.ee/gprofiler/gost) [82], bioDBnet (https://biodbnet-abcc.ncifcrf.gov/db/db2db.php) [83] and UniProt ID mapping (https://www.uniprot.org/uploadlists/). Orthology conversion of mouse-to-human protein accessions was carried out using g:Orth function of g:Profiler and Homologene (https://www.ncbi.nlm.nih.gov/homologene) [84]. We compared proteomes of resting primary CD4+ T cells and SUP-T1 cells from the current dataset with a previously published proteome profile of Jurkat cells [36]. Hypergeometric enrichment-based gene ontology and pathway analysis were carried out with R (R studio v. 1.2.1335, Bioconductor v. 3.9.0) scripts using clusterProfiler (v. 3.12.0).

The datasets were subjected to z-score-based normalization using the scale function of base R (v. 3.6.0) and merged to create matrices. The datasets were then subjected to quantile normalization using normalizeBetweenArrays feature of limma (v. 3.40.6) to account for data distribution skewness between multiple datasets.

### 4.8. Data Availability

Mass spectrometry-derived raw data were deposited to the ProteomeXchange Consortium (http://proteomecentral.proteomexchange.org) via the PRIDE partner repository [85,86]. The data can be accessed using the dataset identifiers PXD015872 for CD4+ T cell data and PXD021272 for SUP-T1 cell data.

## 5. Conclusions

The current study provides a new high-resolution proteomic snapshot of resting CD4+ T cells and after 72 h of activation by TCR, together with a comparison to the proteome of T cell lines and resting/activated human/mouse CD4+ T cells. We confirmed several known T-cell activation-related processes such as IL-2 response, metabolic and signaling changes, cell cycle induction, differentiation into effector cells, among others. The current dataset also provides a resource on checkpoint molecule expression (stimulatory/inhibitory) at this differentiation stage and implicates some proteins such as RIOK2 that previously have not been associated with CD4+ T cell activation. Thus, the data generated from our study may contribute to a better understanding of the proteome transformations in primary CD4+ T cells during T cell activation and the comparability of the proteomes of primary human CD4+ T cells with T cell lines or mouse T cells. The data from our study here, together with other studies, may provide a foundation for developing therapeutic approaches to modulate CD4+ T cell functions.

## Figures and Tables

**Figure 1 ijms-22-00275-f001:**
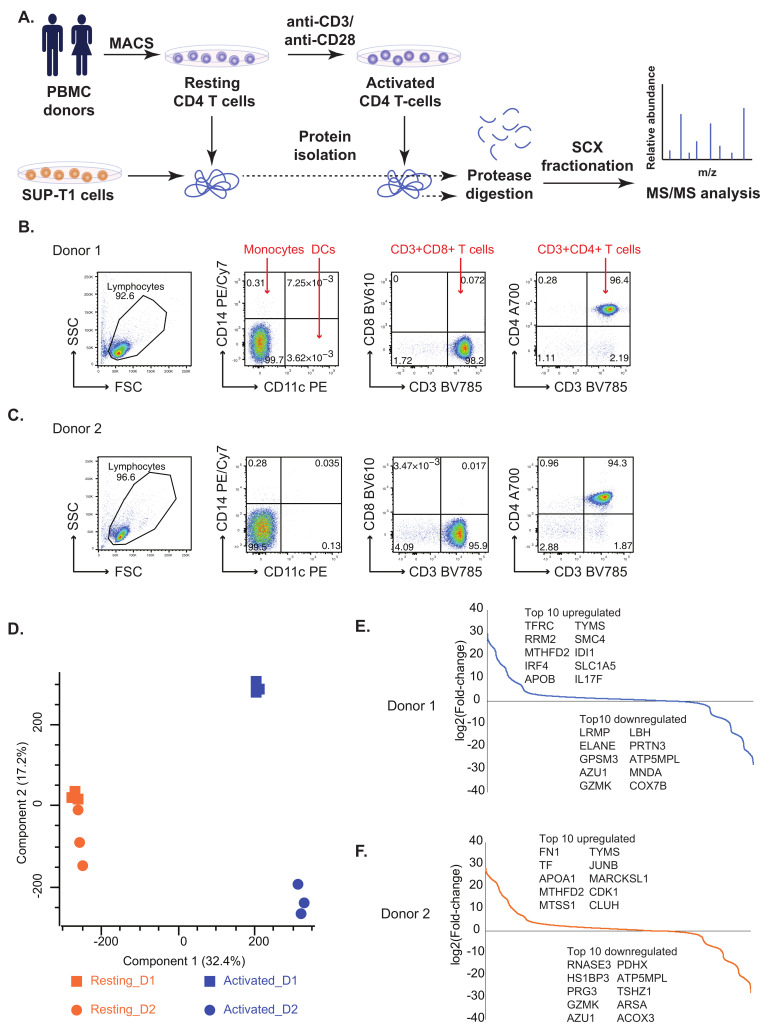
(**A**) A workflow for the comparative proteomics analysis of resting and activated primary human CD4+ T cells as well as SUP-T1 lymphoma T cells. Primary human CD4+ T cells were purified using magnetic beads (MACS cell separation) from PMBCs of healthy donors. Resting (unstimulated) CD4+ T cell samples were harvested (washed and shock-frozen) for protein extraction immediately after isolation. A fraction (1 × 10^7^) CD4+ T cells were activated for 72 h by anti-CD3/anti-CD28 stimulation before cells were harvested/shock-frozen for protein isolation. SUP-T1 cells were harvested from a cell line tissue culture. Resting CD4+ T cells, activated CD4+ T cells and SUP-T1 cell samples were subjected to protein extraction. Proteins were subjected to tryptic digestion. Peptide samples obtained were subjected to strong cation exchange chromatography followed by MS/MS analysis. (**B**,**C**) Flow cytometric purity analysis of CD4+ T cell preparations for Donor 1 (**B**) and Donor 2 (**C**). Purified CD4+ T cells were stained with fluorescent antibodies for CD11c, CD14, CD3, CD8 and CD4. Both preparations contained >94% CD3+CD4+ T cells and (in total) less than 0.5% contaminating CD11c+, CD14+ or CD8+ cells. (**D**) Principal Component Analysis (PCA) plot depicting common proteomic patterns in resting/resting CD4+ T cells and activated CD4+ T cells. (**E**,**F**) S-curve graphs showing the distribution of fold-changes in Donors 1 and 2 and the top differentially expressed proteins.

**Figure 2 ijms-22-00275-f002:**
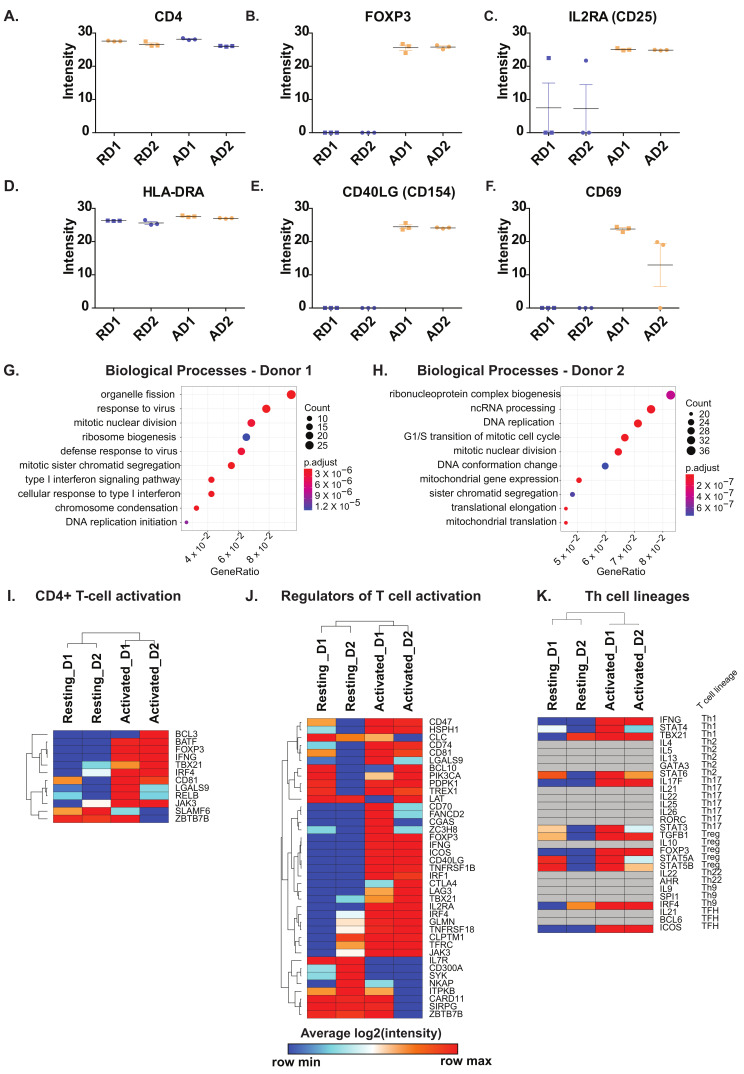
Protein expression levels of (**A**) CD4 (**B**) FOXP3 (**C**) IL2RA (CD25), (**D**) HLA-DRA, (**E**) CD40LG, (**F**) CD69 in resting (R) and activated (A) primary CD4+ T cells from Donor1 (D1) and Donor2 (D2). Enriched Biological Processes from proteins upregulated in (**G**) Donor 1 and (**H**) Donor 2 in response to activation. Heatmaps depicting (**I**) CD4+ T cell activation markers, (**J**) Regulators of T cell activation in CD4+ T cells and (**K**) Markers of T cell lineages. Genesets for comparison of significantly changing proteins were obtained from MSigDB.

**Figure 3 ijms-22-00275-f003:**
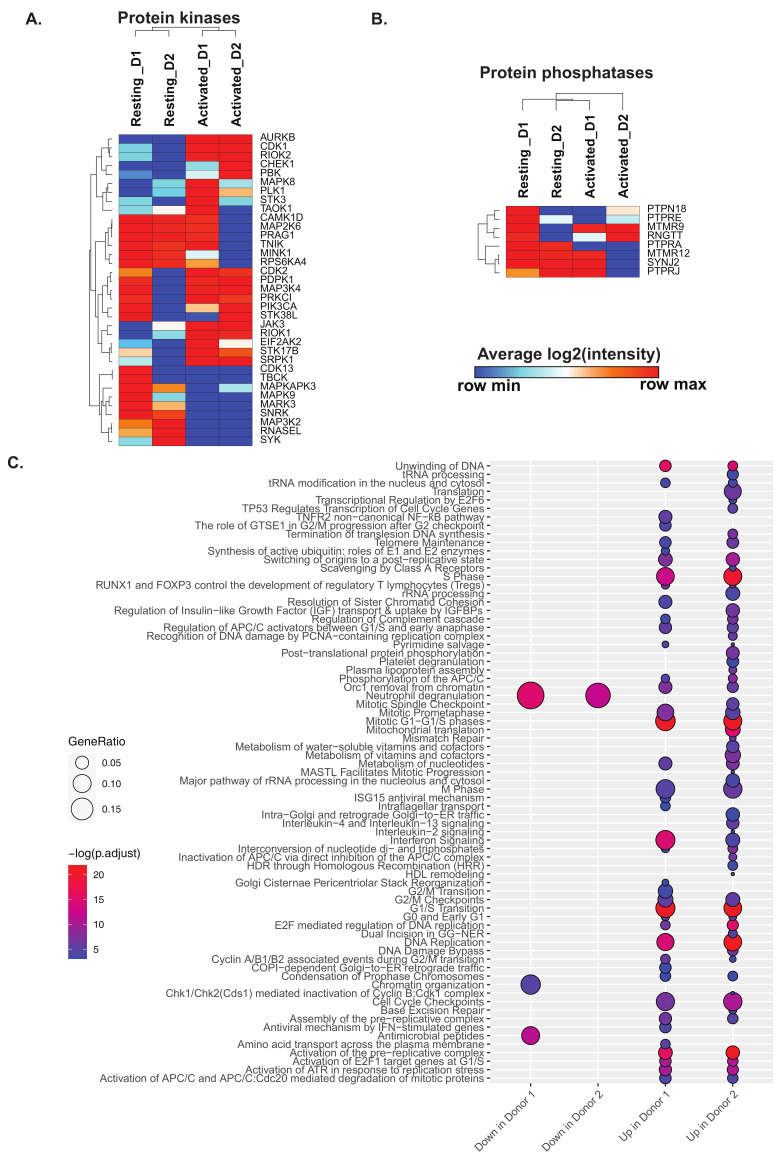
Heatmaps depicting changes in (**A**) protein kinases and (**B**) protein in CD4+ T cells in response to activation. Genesets for comparison of significantly changing proteins were obtained from MSigDB. (**C**) Significantly changing signaling pathways in resting and activated CD4+ T cells after pathway enrichment using Reactome Pathways.

**Figure 4 ijms-22-00275-f004:**
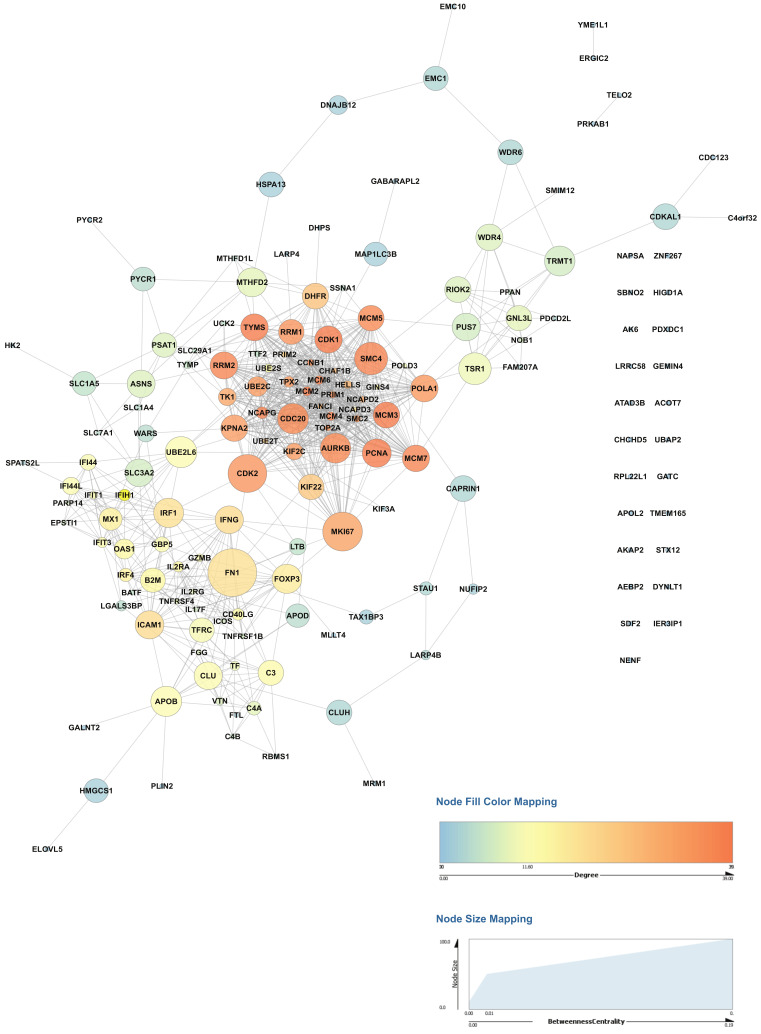
Network analysis of proteins upregulated in primary activated CD4+ T cells in both donors. The Network analysis was carried out using Cytoscape, and network topology properties were calculated using NetworkAnalyzer. The betweenness centrality and degree measures were used to visualize the relationship between nodes. Larger sizes of nodes from high betweenness centrality suggest potential regulatory hubs.

**Figure 5 ijms-22-00275-f005:**
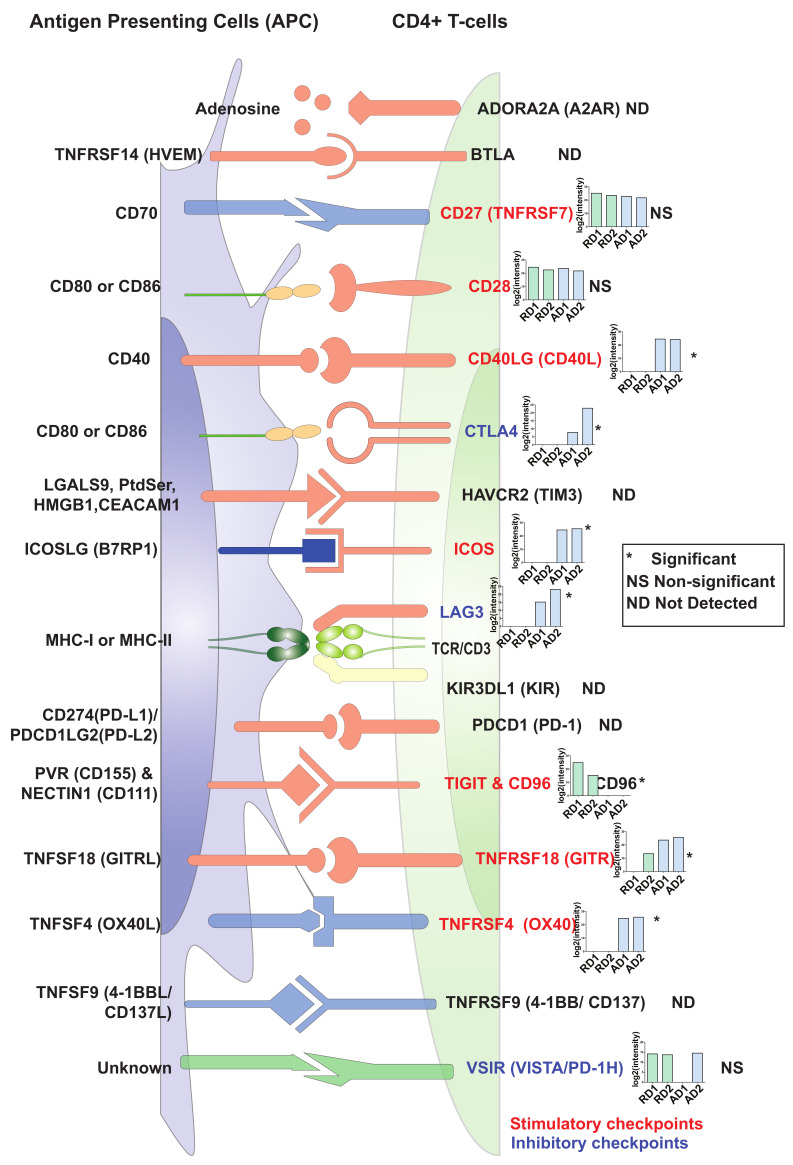
An illustration representing expression profiles of immune checkpoint regulators in resting and activated primary CD4+ T cells. The inset graphs for each protein provide log2(intensity) value-based abundances for resting (R) and CD3/CD28-activated (A) CD4+ T cells from Donor1 (D1) and Donor2 (D2). Legends indicate Significant (*, FDR ≤ 0.05), Not significant (NS), Not detected (ND).

**Figure 6 ijms-22-00275-f006:**
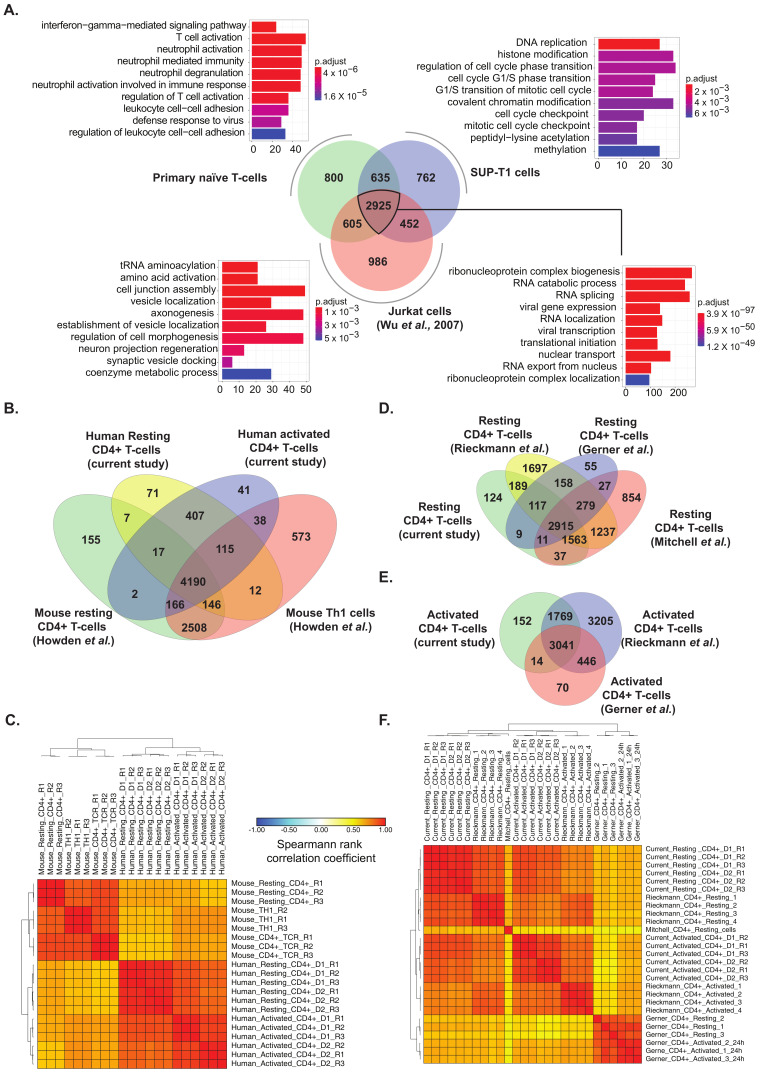
(**A**) Comparison between proteomic profiles of primary resting CD4+ T cells, SUP-T1 cells and Jurkat cells (Wu et al., 2007) and Gene ontology-based classification of biological processes of common proteins and proteins distinct to each cell type (**B**) Comparison of identified proteins and (**C**). Correlation matrix between previously published dataset on mouse (Howden et al.) and human CD4+ T cells (current study). Comparison of lists of identified proteins with previous studies on (**D**) Resting and (**E**) Activated human CD4+ T cells. (**F**) Correlation matrix for proteomic profiles of resting and activated CD4+ T cells from the current study, and previously published studies including Rieckmann et al., Gerner et al., and Mitchell et al.

## Data Availability

Mass spectrometry-derived raw data were deposited to the ProteomeXchange Consortium (http://proteomecentral.proteomexchange.org) via the PRIDE partner repository. The data can be accessed using the dataset identifiers PXD015872 for CD4+ T cell data and PXD021272 for SUP-T1 cell data.

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
