# Peer review of "The Proteomic Landscape of Resting and Activated CD4+ T Cells Reveal Insights into Cell Differentiation and Function"

_ijms, 2020, doi:10.3390/ijms22010275_

Round 1

Reviewer 1 Report

I would like to congratulate to the Authors for this paper. It is extremely interesting, the use of several methods is impressive and the results are very valuable.My only suggestion is to rethink the necessity of showing the results of the well known markers. The Authors are surely aware of the fact that some of the shown results are nothing new and they comment on this in the paper, but in my understanding on the paper, it would have been more novel and of a higher importance and originality if the known facts is removed. Nevertheless, my overall opinion of the paper in very high and I recommend publishing it.

Author Response

We thank the reviewer for the comments.

As per the suggestions of Reviewer 1, we have now revised the manuscript to highlight the novel findings of the study. We have condensed several paragraphs of the Results and Discussion sections of the manuscript as highlighted using “Track changes” to better depict the current study's novel findings. The sections on cell-specific markers that validate the cell types used in the study have been condensed. We hope that the manuscript has been improved significantly now and reads better.

Reviewer 2 Report

The manuscript by Subbannayya et al describes the protein profile of resting and activated CD4+ T cells. Authors analysed several proteins and pathways which are differently regulated in resting versus activated CD4+ T cells. This manuscript contains a large body of work which is explained in depth. The main strength of this study is that primary human cells were used for the analysis which offers a direct translational approach. However this study is limited by the novelty and just covers the phenotypic analysis obtained from proteomic data and there is no functional relevance analysed. Authors claim that they identified some novel proteins such as RIOK2 which may regulate CD4+ T cell functions but not validation is provided. For the novelty and to increase the impact of this study, authors must validate at least one novel protein at functional level.  

Author Response

We thank the reviewer for their positive comments.

We thank the reviewer for the useful suggestions.  We have now revised the manuscript to highlight the novel findings of the study. We have condensed several paragraphs of the Results and Discussion sections of the manuscript as highlighted using “Track changes” to better depict the novel findings in the current study. We hope that the manuscript has been improved significantly now and reads better.

We agree with the reviewer that validating our findings would significantly increase the impact of this study. However, we are currently limited by dedicated funding for this work. We hope to obtain funds for in-depth validation of several poorly characterized nonetheless interesting candidates identified in this study. We plan to validate the role of RIOK2 (RIO kinase 2) in T cell activation when we obtain dedicated funding for the work. In addition, RIOK2 is not a well-studied protein (PubMed searches for RIOK2 yielded 34 results, of which 22 pertain to humans). Though antibodies and knockout mice models are commercially available for the protein, there are no studies that validate the efficacy of these antibodies, and therefore, validation of this particular protein will require substantial effort. Therefore, we currently have explored the role of novel proteins such as RIOK2 using co-expression analysis and anecdotal evidence available in the literature.

From previous literature, we have identified that Polo-like kinase 1 (PLK1) has an effect on T cell function (1,2). In an unrelated study, it was shown that a mechanism where PLK1 phosphorylates and activates RIOK2 was responsible for mitotic progression (3). From these studies, we thus hypothesize that RIOK2 might be involved in the division and function of T cells function and point this out in the revised Discussion of the manuscript. Further evidence using co-expression data from STRINGdb (Figure 1) shows that these proteins are localized in proximity and are likely to be interacting partners.

Figure 1: Protein-protein interaction data from STRINGdb showing that RIOK2 and PLK1 are coexpressed (edge colored in yellow between them).

References

  1. Bostik P, Dodd GL, Villinger F, Mayne AE, Ansari AA. Dysregulation of the polo-like kinase pathway in CD4+ T cells is characteristic of pathogenic simian immunodeficiency virus infection. J Virol. 2004 Feb;78(3):1464-72. doi: 10.1128/jvi.78.3.1464-1472.2004.
  2. Raab M, Strebhardt K, Rudd CE. Immune adaptor SKAP1 acts a scaffold for Polo-like kinase 1 (PLK1) for the optimal cell cycling of T-cells. Sci Rep. 2019 Jul 18;9(1):10462. doi: 10.1038/s41598-019-45627-9. 
  3. Liu T, Deng M, Li J, Tong X, Wei Q, Ye X. Phosphorylation of right open reading frame 2 (Rio2) protein kinase by polo-like kinase 1 regulates mitotic progression. J Biol Chem. 2011 Oct 21;286(42):36352-60. doi: 10.1074/jbc.M111.250175.

Round 2

Reviewer 2 Report

Thank you for providing the reasonable response, further improving and emphasizing the novelty of the manuscript.